# A highly potent human antibody neutralizing all serotypes of BK polyomavirus

Marcel Weber[1], Simone Schmitt[1], Barbara Eicher[1], Jemima Seidenberg[1], Justina Rutkauskaite[1], Benedikt Stöckli[1], Catherine Townsend[1], Uyen Huynh-Do[2], Thomas Schachtner[3], Serena Delbue[4], Armin Mäder[1], Christoph Esslinger [1], Matthias Hillenbrand [1]*

1 Memo Therapeutics AG, Schlieren Zurich, Zurich, Switzerland, 2 Division of Nephrology and Hypertension, University Hospital Bern, Bern, Switzerland, 3 Division of Nephrology, University Hospital Zurich, Zürich, Switzerland, 4 Department of Biomedical, Surgical and Dental Sciences, University of Milan, Milan, Italy

* matthias.hillenbrand@memo-therapeutics.com

## Abstract

BK polyomavirus infection poses a significant risk to kidney transplant recipients. Reactivation of dormant virus in the transplanted kidney, triggered by immunosuppression, can lead to BK polyomavirus-associated nephropathy in up to 10% of transplants, often resulting in loss of graft function or even graft loss. Currently, there is no specific treatment that reliably prevents BK polyomavirus-associated nephropathy or halts its progression. Standard of care relies on reducing immunosuppression, allowing the immune system to gradually control the infection but at the risk of provoking rejection episodes and compromising kidney function. This study describes the discovery and characterization of a highly-potent BK polyomavirus-neutralizing antibody, identified from a kidney transplant recipient who displayed rapid clearance of high-level BKPyV-DNAemia. Antibody mAb 319C07 (USAN potravitug) neutralizes all four BK polyomavirus serotypes and recognizes an epitope critical for viral cell attachment. mAb 319C07 is in clinical development for the treatment of BKPyV infection in renal transplant patients (NCT05769582).

## Author summary

BK polyomavirus (BKPyV) is a common virus that infects most people during childhood and typically remains dormant in the kidneys without causing harm. However, in kidney transplant patients, the immune suppression needed to prevent organ rejection can reactivate the virus. This reactivation occurs in up to 40% of transplant recipients and can lead to BKPyV-associated nephropathy, a serious condition that threatens kidney function and transplant success. Current treatments, such as reducing immunosuppressive drugs, carry risks of organ rejection, and antiviral drugs have shown limited effectiveness. While the role

**Data availability statement:** The structure has been deposited in the RCSB Protein Data Bank (PDB 9RM2). https://www.rcsb.org/structure/9RM2

**Funding:** This work was supported by Innosuisse (18559.1 PFLS-LS to CE). The funders had no role in study design, data collection and analysis, decision to publish, or preparation of the manuscript.

**Competing interests:** I have read the journal's policy and the authors of this manuscript have the following competing interests: MW, SiS, CE, BE, JS, JR, BS, CT, MH and AM, are present or past employees of Memo Therapeutics AG. MW, SiS, CE, TS, UH are inventors on a patent describing mAb 319C07. MW, SiS, CE, BE, JS, JR, CT, MH and AM own options and/or stock of Memo Therapeutics AG.

of antibodies in controlling BKPyV has been debated, recent studies suggest that neutralizing antibodies may help reduce viral levels in the body. Using our powerful antibody discovery platform, we identified several promising antibodies from healthy individuals and transplant patients. One outstanding candidate, mAb 319C07, showed strong activity against all BKPyV serotypes and targets a key viral protein needed for infection. This antibody is now in clinical development and may offer a safer, more effective way to treat or prevent BKPyV-related complications in transplant patients.

## Introduction

BK polyomavirus (BKPyV) is a polyomavirus with a double-stranded, circular DNA genome of approximately 5,000 base pairs [1]. Its major capsid protein, VP1, forms a non-enveloped capsid comprising 72 VP1 pentamers in which other viral proteins, VP2 and VP3, are incorporated. VP1 is the only protein exposed at the outer surface of the capsid. Four major serotypes (I, II, III, and IV) of BKPyV VP1 are distinguished, each engaging sialylated glycans in the cell membrane for cell entry [1,2]. Major sites of BKPyV infection are renal tubular epithelial cells and bladder epithelial cells [3].

BKPyV infection usually occurs during childhood and remains latent in renal tubular cells and the urothelium typically with no or minimal clinical consequence [3]. It is estimated that up to 90% of the adult population is infected [4]).

In kidney transplant recipients, immunosuppression frequently reactivates latent BKPyV, usually in the transplanted kidney [5,6]. Up to 40% of kidney transplant recipients experience BKPyV reactivation [6], and up to 70% of the viremic patients develop BK polyomavirus-associated nephropathy (BKPyV-nephropathy) [6–10]. BKPyV-nephropathy can severely impact graft function and graft longevity, potentially leading to graft failure [6,11]. Currently, the standard of care—reducing immunosuppressive therapy—carries a substantial risk of acute or chronic graft rejection [6,12–14].

Investigations into antiviral agents such as cidofovir, leflunomide, and quinolones have produced inconclusive data regarding potential benefits [6]. Adoptive transfer of virus-specific allogeneic T cells has shown only limited efficacy, with modest reductions in BKPyV-DNAemia [15].

The role of humoral immunity in preventing BKPyV infection has long been a matter of debate in numerous small clinical studies, and intravenous immunoglobulins have historically served as a last-resort option, albeit with limited, or only indirect evidence of benefit [6,16,17].

More recent studies, however, may suggest a protective effect of antibodies. Fafi-Kremer and colleagues describe that pre-existing neutralizing antibody titers correlated with reduced post-transplant BKPyV-DNAemia. Furthermore, they observed that rising neutralizing antibody titers in patients with viruria, viremia, or BKPyV nephropathy coincided with declining viral loads in urine and/or blood [18].

A candidate therapeutic monoclonal antibody, MAU868, demonstrated moderate efficacy in a phase II study (NCT04294472). After 16 weeks of treatment, BKPyV-DNAemia was reduced by at least 1 log in 40% of patients [19]. These findings suggest that antibodies could be a viable approach to reduce viral load and potentially prevent progression to BKPyV-nephropathy.

Leveraging our powerful antibody discovery platform DROPZYLLA, previously used for fast-track discovery of SARS-CoV-2-neutralizing antibodies [20], we comprehensively assessed the BKPyV-neutralizing antibody repertoires in healthy donors and kidney transplant recipients who clinically demonstrated fast viral clearance.

Using this approach, we identified nine antibodies with equal or superior properties to a benchmark BKPyV-neutralizing antibody P8D11. One candidate, mAb 319C07, with high potency against all BKPyV-serotypes and targeting a key epitope on VP1 required for cell entry, was selected. mAb 319C07 (aliases AntiBKV, potravitug) is currently under clinical development for the treatment of BKPyV infection in kidney transplant patients (US FDA Fast Track, NCT05769582) [21].

## Results

### Antibody discovery strategy

We recruited more than forty healthy donors and over thirty kidney transplant recipients (KTR) to maximize the diversity of BK polyomavirus-specific immune responses (Fig 1). Healthy donors were selected based on the presence of measurable neutralizing antibody titers, whereas KTR participants were additionally required to demonstrate rapid viral clearance.

Antibody discovery employed Memo Therapeutics' DROPZYLLA platform [20], which integrates droplet microfluidics to capture antibody genes from up to one million single human peripheral blood memory B cells (typically up to 200,000 memory B cells were obtained per sample). Cognate heavy- and light-chain variable sequences remain linked, yielding expression libraries of full-length, membrane-bound IgG which were expressed in HEK293T cells. We screened these libraries via fluorescence-activated cell sorting (FACS) using fluorescently labeled pentameric VP1 capsid protein (Fig 1). Subsequent selection steps included binding studies against VP1 proteins of all four BKPyV serotypes and cell-based neutralization assays utilizing a well-established pseudovirus system (Pastrana/Buck). This approach identified 29 antibodies demonstrating strong neutralization across all BKPyV serotypes. To select the lead candidates, we benchmarked these 29 antibodies against mAb P8D11, a human-derived antibody with picomolar affinity and potent neutralization activity [22]. The antibody mAb P8D11 was subsequently developed into the clinical candidate MAU868 for the treatment of BKPyV infection in kidney transplant recipients. Nine antibodies (7 derived from kidney transplant recipients) were

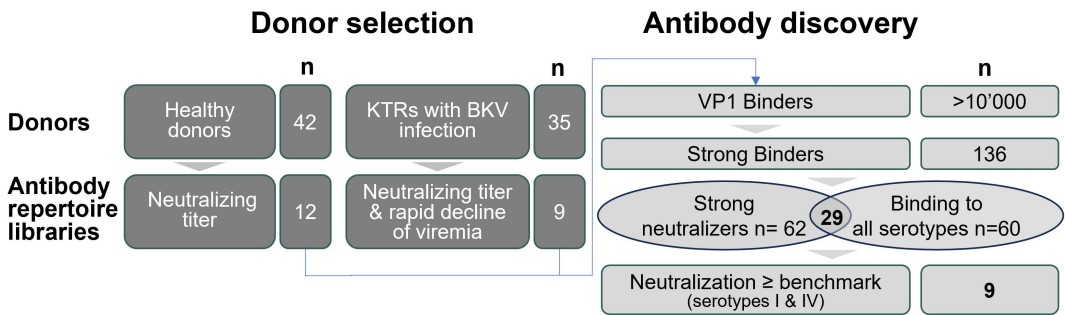

**Fig 1. Antibody discovery was performed using the DROPZYLLA platform, which allows for the cloning and preservation of cognate heavy and light chain pairing of single B cells.** Antibody repertoire libraries were expressed as membrane IgG in mammalian cells and cells that bound BKPyV VP1 pentamer were isolated using antigen-specific sorting. Antibody specificity was confirmed semi-quantitatively using soluble IgG, followed by cell-based neutralization and cross-reactivity testing against the four BKPyV serotypes. The neutralization potency of pan-serotype neutralizing antibodies was then compared to the benchmark antibody P8D11, and a development candidate, mAb 319C07, was selected based on neutralization efficacy and developability.

 

prioritized and evaluated for developability, including expression efficiency, stability, purity, and sequence liabilities. Based on neutralization potency across all serotypes and favorable developability profiles, mAb 319C07 was selected as the development candidate.

mAb 319C07 is a human IgG1/kappa antibody isolated from a cognate memory B cell repertoire library obtained from peripheral blood of a kidney transplant recipient who exhibited rapidly increasing BKPyV-DNAemia at 5 months post-transplant, surpassing the threshold of 10,000 IU/mL (International Units/mL determined by the cobas BKV test, Roche Diagnostics) within two weeks of initial detection. In response, immunosuppressive therapy was adjusted by reducing doses of both the antiproliferative agent and the calcineurin inhibitor. Subsequently, BKPyV-DNAemia continued to rise, peaking at 190,000 IU/mL less than one month later, but then sharply decreased to 90,000 IU/mL within the following five days. It was at this critical juncture that the blood sample was collected for antibody repertoire analysis. Following sample collection, BKPyV-DNAemia further declined, eventually becoming undetectable around nine months after reaching the peak.

## mAb 319C07 binds to all major serotypes of BKPyV with picomolar affinity

Binding of mAb 317C09 to BKPyV VP1 serotypes I, II, III and IV was analyzed by ELISA and showed picomolar affinity of mAb 319C07 to all four serotypes (Fig 2 and Table 1). Benchmark antibody mAb P8D11 showed similar affinity for

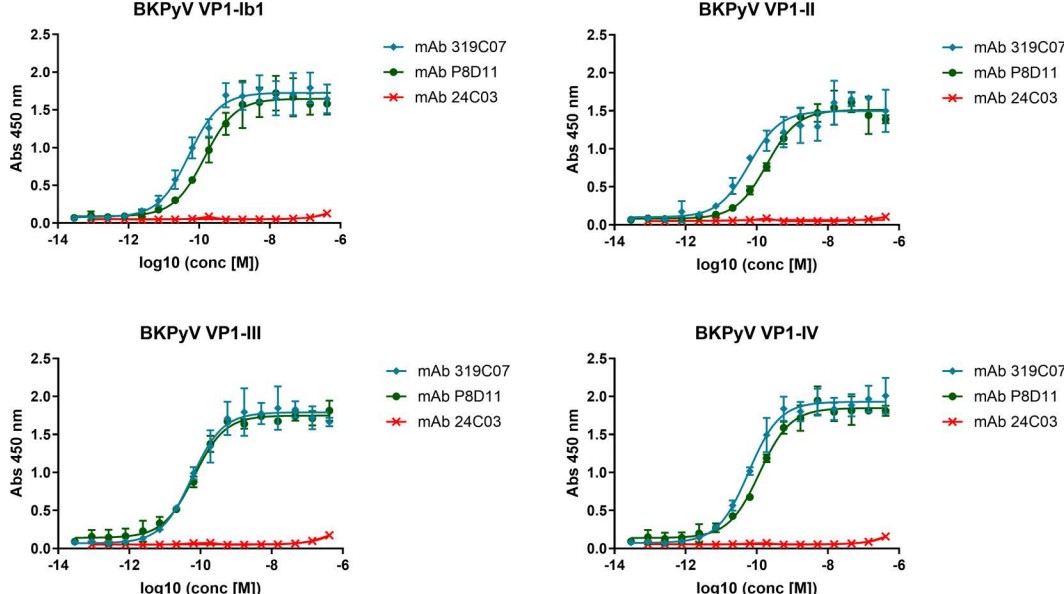

**Fig 2. Binding of mAb 319C07, mAb P8D11 and mAb 24C03 (negative control) to BKPyV VP1-Ib1, II, III or IV assessed by ELISA.** The mean of triplicate wells ±SD from a representative experiment was plotted, and nonlinear sigmoidal dose-response regression was used to fit the curves for each mAb. EC50 values for mAb319C07 and mAb P8D11 are shown in Table 1.

**Table 1. Summary of EC50 values (in nM) determined by ELISA.**

|  | EC50 (95% CI) in (nM) | |
| --- | --- | --- |
| VP1 SEROTYPE | mAb 319C07 | mAb P8D11 |
| Ib1 | 0.05 (0.04-0.07) nM | 0.14 (0.11-0.20) nM |
| II | 0.06 (0.04-0.10) nM | 0.19 (0.15-0.25) nM |
| III | 0.06 (0.04-0.08) nM | 0.07 (0.05-0.08) nM |
| IV | 0.06 (0.05-0.07) nM | 0.12 (0.10-0.15) nM |

serotype III, but slightly decreased affinity for serotypes I, II and IV. mAb 24C03, an antibody directed against tetanus toxoid, was used as a negative control and did not show binding to VP1.

## Inhibition of BKPyV-VP1 binding to target cells by mAb 319C07

We investigated whether mAb 319C07 could block the binding of BKPyV VP1 pentamer to host cells. In a flow cytometry assay using HEK293TT cells as target, VP1 pentamer attachment to cells was measured in the presence of increasing antibody concentrations. mAb 319C07 and mAb P8D11 showed a dose-dependent inhibition of cell attachment for all VP1 serotypes, although mAb P8D11 required higher concentrations to show this effect (Figs 3 and S1 and Tables 2 and S1). These results suggest that the neutralizing effect of both antibodies is likely mediated via interference with the sialylated glycan-binding step in the infectious entry process. Control antibody mAb 24C03 did not show any effect on VP1 binding.

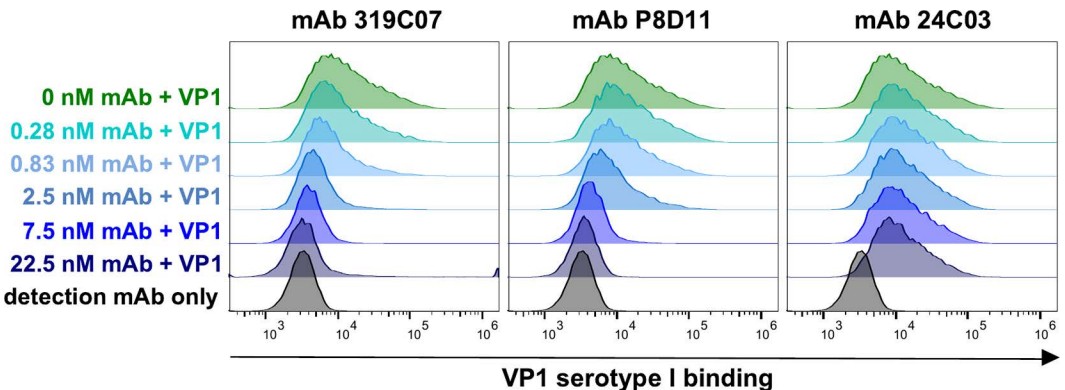

**Fig 3. Inhibition of binding of BKPyV VP1 serotype Ib1 to HEK293TT cells in the presence of mAb 319C07.** Attachment of VP1 serotype Ib1 pentamer to HEK293TT cells in the absence (dark green) or presence of antibody (mAb319C07, mAb P8D11 or mAb 24C03) at different concentrations (shades of blue) was analyzed. VP1 binding to HEK293TT cells was detected using an AF488 labelled anti-VP1 antibody and compared to cells with detection antibody only (black). Analysis was done by analytic flow cytometry. A shift of the HEK293TT population to higher signals (right-shift) over background (black) indicates binding of VP1 to the cells.

**Table 2. Percent binding of BKPyV VP1 serotype Ib1 to HEK293TT cells in the presence or absence of antibodies.** Median fluorescent intensity of plots shown in Fig 3 were extracted using FlowJo 10.7.1 and normalized to 100% VP1 binding (VP1-Ib1 binding to HEK293TT cells in the absence of antibodies) and 0% VP1 binding (HEK293TT cells with detection antibody only). For mAb 319C07 and mAb P8D11 a concentration-dependent decrease in VP1 binding to the cells was observed. mAb 24C03 does not inhibit VP1 binding to HEK293TT cells. n = 1, 15000 cells analyzed per concentration.

| | VP1 Ib1 binding to HEK293TT cells | | |
|---|---|---|---|
| Antibody (nM) | mAb 319C07 | mAb P8D11 | mAb 24C03 |
| 22.5 | 1% | 3% | 100% |
| 7.5 | 8% | 14% | 104% |
| 2.5 | 19% | 53% | 110% |
| 0.83 | 47% | 95% | 126% |
| 0.28 | 72% | 116% | 121% |
| 0 | 100% | 100% | 100% |

**Table 3.** In vitro neutralizing activity of mAb 319C07 and mAb P8D11 against the 4 major BKPyV serotypes. Neutralization activity was determined in HEK293TT cells by measuring luciferase expression 72 hours after infection with pseudoviruses. Data are expressed as the median IC50 ± SD from 3 (serotypes I, II, and III) or 10 (serotype IV) independent experiments.

| | IC50 (pM) | |
|---|---|---|
| BKPyV SEROTYPE | mAb 319C07 | mAb P8D11 |
| I | 0.11 ± 0.06 | 7.62 ± 2.3 |
| II | 0.35 ± 0.05 | 11.13 ± 0.29 |
| III | 29.53 ± 6.16 | 6.71 ± 1.25 |
| IV | 0.88 ± 1.15 | 20.625 ± 10.03 |

### Neutralization of All BKPyV Serotypes by mAb 319C07

To determine the neutralization potency of mAb 319C07, a cell-based pseudovirus assay was employed. The pseudoviruses comprised BKPyV structural proteins VP1, VP2, and VP3, packaging a luciferase reporter gene as payload. These virus-like particles authentically mimic BKPyV capsid structure and immunogenic properties, enabling accurate quantification of infection via luciferase activity.

In these assays, antibody mAb 319C07 exhibited robust neutralization across all four major BKPyV serotypes (I–IV) and significantly outperformed benchmark antibody mAb P8D11 against serotypes I, II, and IV (Fig 4), which together account for over 98% of global BKPyV infections [23].

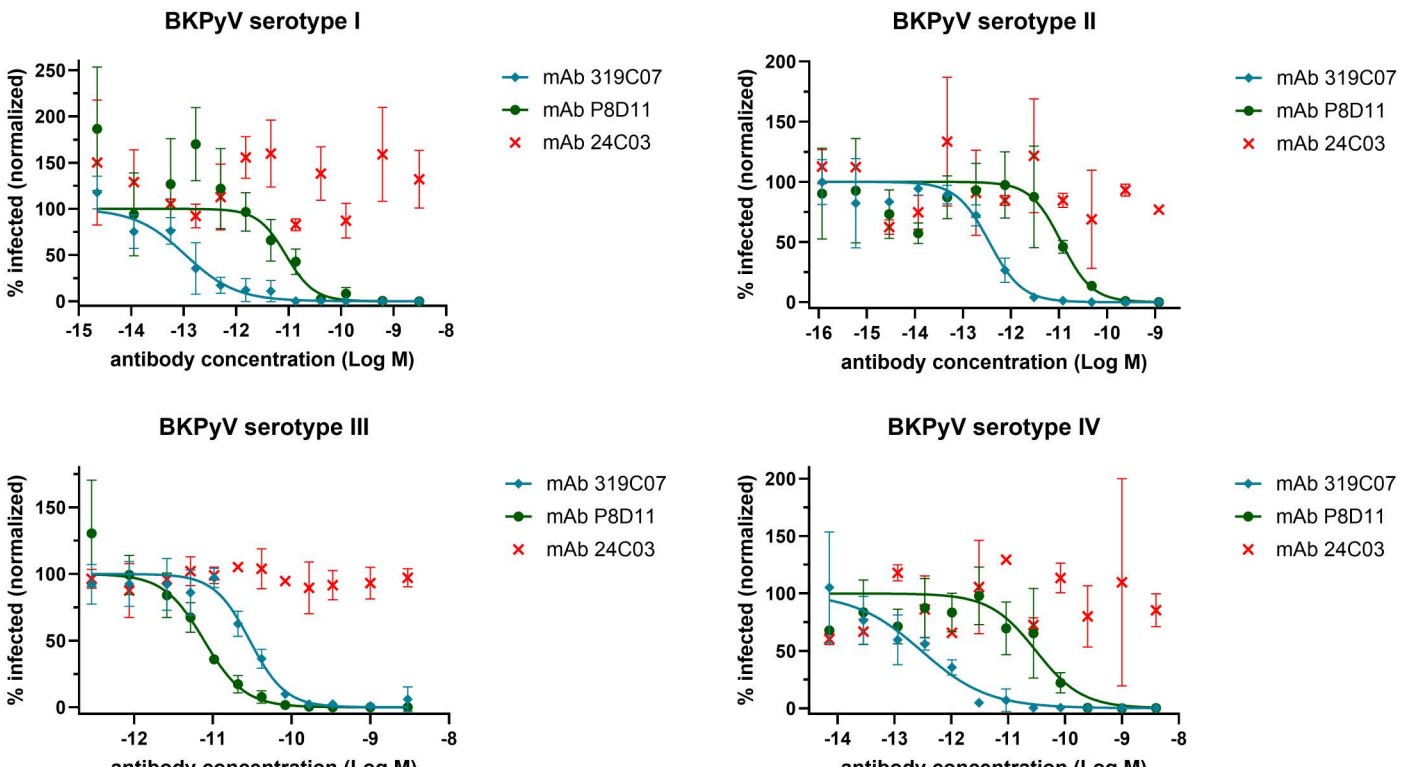

**Fig 4. Neutralisation of BK polyoma (BKPy) pseudovirus serotype I, II, III and IV by mAb 319C07 and mAb P8D11.** Control mAb 24C03 is directed against tetanus toxoid and does not show neutralization of BKPyV. The mean of triplicate wells ±SD (except mAb 24C03 which was done in duplicate only) from a representative experiment was plotted, and nonlinear sigmoidal dose-response regression was used to fit the curves for each mAb. IC50 values for mAb319C07 and mAb P8D11 are shown in Table 3.

Representative IC50 values of mAb 319C07-neutralization are also summarized in Table 3.

We next compared the neutralization potency of mAb 319C07 to that of a commercially available immunoglobulin preparation. Immunoglobulin preparations are commonly administered as an adjuvant therapy in kidney transplant recipients exhibiting insufficient viral clearance following immunosuppression reduction. Our findings demonstrate that the neutralization potency of the immunoglobulin preparation is approximately four logs lower than that of mAb 319C07 (Fig 5). Considering typical administered doses—immunoglobulin at 20–140 g per injection versus potravitug (mAb 319C07) at 1 g per injection—mAb 319C07 is estimated to provide approximately 100–500-fold greater neutralizing capacity compared to the immunoglobulin.

## BKPyV VP1 epitope determination for antibody mAb 319C07

For epitope determination, protein crystallography was used to solve the structure of Fab fragments of mAb 319C07 bound to BKPyV VP1 Ib1 pentamer at a resolution of 1.96 Å. The structure reveals the binding of five Fab fragments of 319C07 to the five protomers of the VP1 pentamer on the capsid outside-facing surface of the pentamer (Fig 6A). An extensive network of interactions is formed between the VP1 pentamer and complementarity-determining regions (CDRs) of heavy (CDR H1 and CDR H3) and light (CDR L1 and CDR L2) chain of mAb 319C07 (Fig 6B). Although most contacts are mediated by the main VP1 protomer, both CDR1 and CDR3 of the heavy chain make contact to the clockwise (cw) neighboring VP1 protomer involving D75cw and F76cw. In addition, the same two CDRs of the heavy chain also contact a sidechain of the counterclockwise (ccw) VP1 protomer (H139ccw). Notably, by forming contacts to VP1 amino acid residues D75cw, F76cw, G132, H139ccw, N273, S274, S275, and T277, the CDR3 of the heavy chain wedges itself into the binding pocket for ganglioside GD3 as reported by Neu et al. [24] (Fig 6C). The structure also offers a rational why potency on serotype III is reduced: in the structure, side chain of K69 on VP1 forms a polar interaction with T57 of the light chain of mAb 319C07, which would not be possible with the H69 side chain present in serotype III.

Overall, the epitope on VP1 extends itself to three distinct regions: the hypervariable BC-loop, the DE loop and the highly conserved HI loop [25,26] (Fig 6D), which are all involved in BKPyV interaction with sialylated glycans. Consequently, mAb 319C07 effectively mimics GD3, sharing a substantial number of key residues involved in VP1 binding to

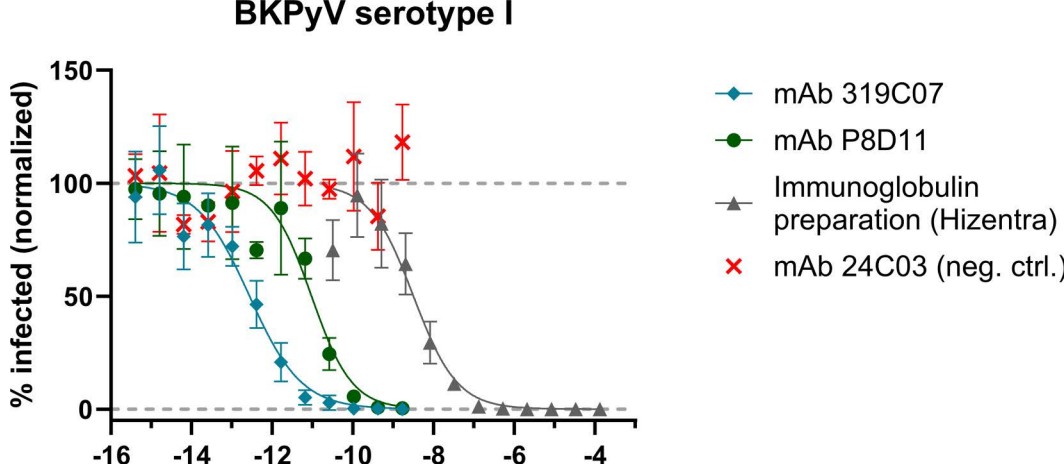

**Fig 5. Comparative analysis of neutralisation potency of monoclonal antibodies 319C07, P8D11 and an immunoglobulin preparation (Hizentra).** Serial dilutions of antibodies were tested in a cell-based infection assay using BKPyV serotype I pseudovirus. The mean values of triplicate wells ± SD from a representative experiment was plotted, and nonlinear sigmoidal dose-response regression was used to fit the curves for each mAb.

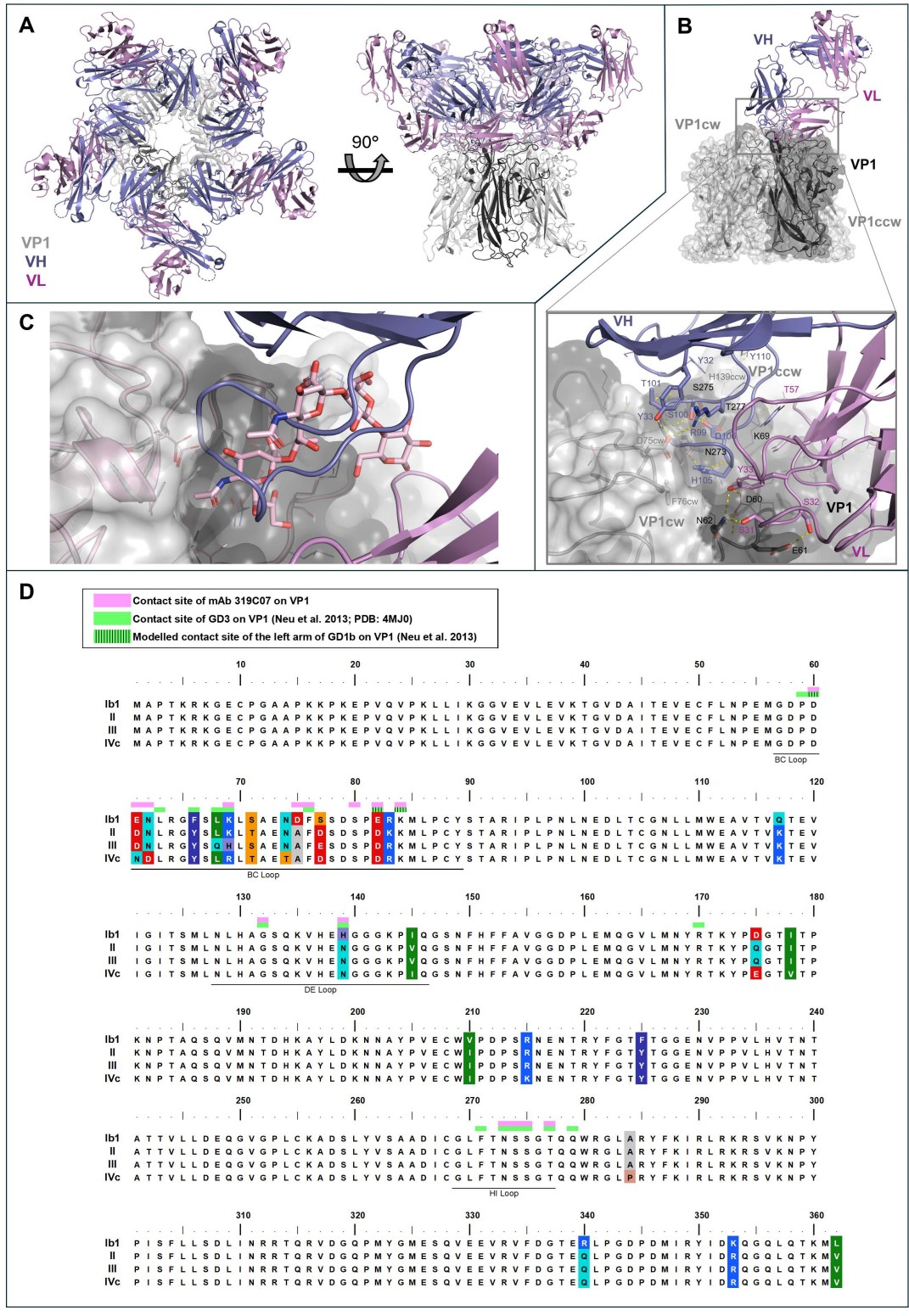

**Fig 6. X-ray crystallography of the complex between the Fab fragments of mAb 319C07 and the BKPyV-VP1 pentamer (PDB 9RM2).** (A) The complex shown from the capsid outside-facing surface (*left*) or from the side (*right*). The heavy (violet) and light (magenta) chain of mAb 319C07 bind the VP1 pentamer (grey) on the capsid outside-facing surface. One of the VP1 protomers is highlighted in black. (B) Similar representation as in (A) but reduced to one Fab fragment and the three VP1 protomers it makes contacts to: the main VP1 in black and the clockwise (cw) and counterclockwise (ccw), both grey. The excerpt provides a zoomed in view on the region of the main interaction between mAb319C07 and VP1 and shows a selection of residues involved in this interaction. Sidechains and labels of the residues shown are depicted in the same color as their corresponding chain. H-bonds are depicted as yellow dashed lines. CDR L1: aa24-35; CDR L2: aa51-57; CDR H1: aa26-32; CDR H3: aa98-110 (C) Overlay of the structure described herein and the structure by Neu et al. [24] (PDB 4MJ1), showing how CDR H3 (violet loop) and ganglioside GD3 (shown in pink stick model) compete for the same binding pocket on VP1. (D) Sequence alignment of VP1 serotypes Ib1, II, III and IVc. Positions with polymorphisms between serotypes are highlighted using the RasMol coloring scheme. BC, DE and HI loops of VP1 are indicated by solid lines. Bars on top of VP1 residues indicate their involvement in interaction with mAb 319C07 (pink), GD3 (green; as reported by Neu et al. [24], PDB 4MJ1) and left arm of GD1b (green with vertical stripes; identified by Neu et al. [24] using modelling).

sialylated glycans, including those contributed by adjacent protomers. This mode of binding may explain how mab319C07 prevents virus cell entry and suggests that viral escape by mutation may be highly unlikely.

The epitope recognized by mAB 319C07 significantly differs from those reported for other BKPyV-neutralizing antibodies, such as MAU868 [22] and 41F17 [27]. Both antibodies were identified through screens for binding to virus-like particles, whereas mAb 319C07 was selected based on VP1 pentamer binding. The MAU868 scFv binds exclusively to a single VP1 monomer within the pentamer, while the 41F17 scFv interacts with three VP1 monomers spanning two adjacent pentamers. In contrast, neither antibody targets the highly conserved DE (amino acid stretch N128–Q146) and HI loop (amino acid stretch N273–T277), which are bound by mAb 319C07 and are essential for sialylated glycan binding and subsequent viral entry into host cells.

## Resistance profiling in long-term culture

The neutralization capacity of mAb 319C07 against BK polyomavirus serotypes I and IV, as well as the potential emergence of viral escape mutants, was assessed using primary human renal proximal tubular epithelial cells (HRPTEC) infected with replication-competent virus [27]. HRPTEC seeded in 96-well plates were incubated with mixtures of BK polyomavirus and antibody. After one week of culture, a portion of the supernatant was harvested for viral quantification by qPCR, while remaining cells underwent freeze-thaw cycles to release virus. For the subsequent cycle, virus containing lysate was added to a fresh HRPTEC-antibody co-culture. This procedure was repeated every two weeks for up to six cycles. Viral copy number was quantified by qPCR and viral DNA, extracted from lysed cells after the final passage was used for genotyping by sequencing of the VP1 region (Fig 7).

Quantitative PCR confirmed an antibody concentration-dependent reduction in BKPyV-DNAemia, with mAb 319C07 demonstrating superior neutralizing activity relative to the benchmark mAb P8D11 (Fig 7A). No escape mutants emerged in this assay, as confirmed by sequence analysis of the VP1 region showing consensus amino acid sequence of VP1 (Fig 7B). These results were reproduced for both serotype I (Fig 7) and serotype IV (S2 Fig) in independent experiments.

## Discussion

BK polyomavirus (BKPyV) infection represents a critical and unmet medical need among kidney transplant recipients, significantly contributing to graft dysfunction and loss [7,11]. The current standard of care is primarily centered on the reduction of immunosuppression. This, in turn, is associated with a substantial risks of graft rejection and compromised kidney function and kidney longevity [6,8]. Consequently, targeted antiviral therapy remains a high unmet medical need. Here, we present the characterization of mAb 319C07 (AntiBKV, potravitug), a novel human monoclonal antibody exhibiting highly potent neutralization across all BKPyV serotypes which is currently in clinical development for the treatment of BKPyV infection in kidney transplant recipients (NCT05769582).

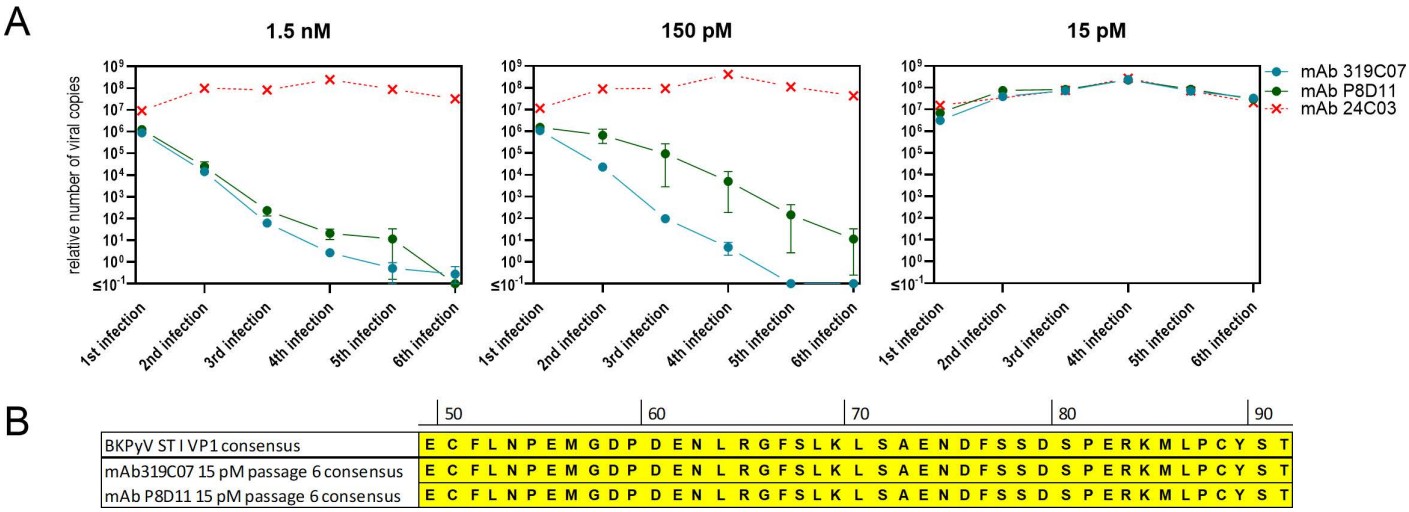

**Fig 7. Long term culture of replication competent BKPyV in the presence of antibody mAb 319C07 and subsequent analysis of putative VP1 sequence alterations.** (A) Neutralization capacity of mAb 319C07 and reference antibody mAb P8D11 plus negative control mAb 24C03 on HRPTEC infected with BKPyV serotype I was quantified. With antibodies mAb 319C07 and reference antibody mAb P8D11 assessment was performed in duplicate at all concentrations, for mAb 24C03, the highest concentration was used. Plotted are mean and ranges of n = 2. (B) Sequence analysis performed with the 6th infection passage showed no occurrence of mutated virus.

The therapeutic potential of neutralizing antibodies against BKPyV has recently been underscored in clinical studies investigating the pre-existing BKPyV-neutralizing antibody titers or aiming at complementing lacking neutralizing titers with the administration of immunoglobulin preparations or a monoclonal antibody.

Fafi-Kremer and colleagues reported that pre-existing neutralizing antibody titers exceeding a defined threshold were associated with reduced BKPyV-DNAemia following transplantation [18]. In a subsequent clinical study by the same group, prophylactic administration of immunoglobulin preparations to patients with baseline neutralizing antibody titers below this protective threshold led to a decrease in BKPyV-DNAemia levels comparable to those observed in patients who initially exhibited high neutralizing antibody titers at transplantation [28]. However, this study was limited by a small sample size (n = 3) in the immunoglobulin-treated group.

Moreover, in a recent phase II clinical trial in kidney transplant recipients infected by BKPyV, the therapeutic monoclonal antibody MAU868 resulted in a more rapid and more pronounced reduction of BKPyV-DNAemia compared to placebo, where the intervention consisted solely of reducing immunosuppressive therapy. After 16 weeks of treatment, BKPyV viremia (DNAemia) decreased by at least 1 log in 40% of patients, with a subset achieving undetectable viral levels — a response not observed in the placebo group [19]. The study did not assess effects on important disease parameters such as BK polyomavirus-associated nephropathy (BKPyV-nephropathy) as no biopsies were taken. Nevertheless, these findings indicate that antibody-based therapies may represent a promising approach to reduce viral loads and potentially prevent progression to BK polyomavirus-associated nephropathy (BKPyV-nephropathy).

In this context, antibody 319C07, which demonstrated substantially greater neutralization potency compared to mAb P8D11 and conventional immunoglobulin preparations, may result in enhanced clinical efficacy.

Mechanistically, antibody 319C07 robustly inhibited BKPyV VP1 pentamer attachment to host cells, indicating that interference with the sialylated glycan-binding step in the infectious entry process is its primary antiviral mechanism. Structural crystallographic analyses further support this by identifying an epitope stretched over three VP1 protomers, where the antibody's CDR H3 loop extends directly into sialylated glycan-binding sites on the viral capsid with the light chain making additional contacts to the BC loop. This interaction likely prevents viral attachment, underpinning the potent antiviral

activity observed in neutralization experiments. The observation that both the heavy and light chains of the antibody mAb 319C07 make critical contacts underscores the importance of preserving native heavy-light chain pairing from source B cells during antibody discovery—a capability provided by the DROPZYLLA platform employed in this study.

Targeting such a critical epitope may increase the potential for viral escape, a well-documented concern in antiviral therapies. However, the high degree of overlap with sites making critical contacts to the ganglioside GD3 -distributed over BC, DE and HI loops- suggests that escape mutations would likely compromise host cell specificity and viral infectivity. Although the clinical relevance of our long-term viral passaging experiments using primary human renal proximal tubular epithelial cells (HRPTEC) will require confirmation in clinical studies, the experiment, adapted from Lindner et al. [27], demonstrated sustained antiviral efficacy of mAb 319C07 without detectable emergence of escape mutants over multiple replication cycles. Interestingly, Lindner et al. reported viral escape mutants as early as after the first passage for some of the antibodies tested in this assay.

Taken together, these findings suggest a low risk of viral escape with antibody mAb 319C07, reducing concerns about diminished clinical efficacy during prolonged therapeutic use. The identification of mAb 319C07 from a kidney transplant recipient who exhibited rapid viral clearance may increase the chance for successful translation into the clinic. The efficient isolation and development of potent human antibodies from such donors using advanced platforms, such as DROPZYLLA, could exemplify a novel paradigm for accelerated therapeutic antibody discovery tailored to highly specific medical challenges.

Looking forward, the exceptional potency, broad serotype coverage, defined mechanism of action, and barrier to resistance may position antibody mAb 319C07 as a transformative therapy for managing BKPyV infections in kidney transplant recipients. The ongoing clinical development will assess whether this approach can complement current therapeutic strategies for BKPyV infection in kidney transplant recipients, potentially reducing the need for immunosuppression minimization and its associated risk of allograft rejection.

## Materials and methods

### Ethics statement

Donor recruitment was performed according to the rules set out by the Cantonal Ethics Commission of Zurich. Approval was granted under the number BASEC-2016–01260, all donors provided formal consent in written form.

### Production of BKPyV VP1 pentamer

For expression, BKPyV VP1 antigens of serotype Ib1, II and IVc2 (aa 31–300) with N-terminal Thrombin site and hexa-His tag were ordered as gene synthesis constructs in a pET15b expression vector. The constructs were used to transform *E. coli* BL21 (DE3) cells by heat shock and a culture grown from a single clone. Protein expression was induced at OD600 of ~0.6 with 0.2 mM IPTG and cells harvested after overnight cultivation by centrifugation. Cells were resuspended in purification buffer (50 mM Tris, 250 mM NaCl, 10 mM imidazole, 5% glycerol pH 7.5) supplemented with 1 mM PMSF and lysed by sonication, followed by centrifugation and filtration to clarify the supernatant. Proteins were purified via affinity chromatography using a NiNTA resin. For eluted proteins buffer was exchanged by dialysis (VP1 Ib1), and followed by a second purification step by size-exclusion chromatography for some of the purified proteins (VP1 II, VP1 IVc2). Purity and integrity of the proteins was confirmed by Coomassie stained SDS-PAGE under reducing and non-reducing conditions. VP1 serotype III was obtained from Abcam (ab74567), where the protein was expressed in *S. cerevisiae* and purified tag-free.

### Antibody discovery

Memory B cells, defined to be the cell population positive for CD22 and negative for CD3, CD8, CD56, IgA, IgD, and IgM, were isolated from peripheral blood of donors and used to prepare cognate antibody repertoire expression libraries as described in patent WO2015121434. Briefly, single memory B cells were lysed in microfluidic droplets. Upon lysis,

the mRNA was captured on mRNA capture beads, and after RT-PCR, single mRNA capture beads with attached cDNA were re-encapsulated in droplets and immunoglobulin heavy and light chain variable domain sequences were amplified by droplet PCR [29,30]. PCR fragments of linked immunoglobulin light chain and heavy chain variable regions were cloned into an expression vector, providing the human immunoglobulin constant heavy region combined with a transmembrane domain derived from human CD8 to allow for mammalian cell display of the antibodies. HEK293T cells displaying membrane-bound full-length IgG (mIgG-HEK) were generated by transduction of HEK293T cells with lentiviruses encoding the antibody repertoire. Screening of the antibody libraries was performed by flow cytometric sorting of antigen-specific cells utilizing a dual-labelling strategy (BKPyV-VP1 pentamer labelled with either APC or PE) as described previously [20]. Single-cell-sorting yielded HEK293T cell clones expressing monoclonal antibodies with reactivity to VP1. Antibody-specificity was confirmed in analytical flow cytometry, testing for high-affinity binding of cell-membrane-expressed antibodies to purified VP1 protein.

### Production of Human Monoclonal Antibodies

Selected antibodies were cloned into expression vectors for soluble IgG1 expression. Sequence information of antibody P8D11 sequence was obtained from patent WO2017046676A1, ordered as a gene synthesis construct and cloned into IgG1 expression construct. Negative control/isotype control antibody (24C03) is a human-derived antibody specific for tetanus toxoid. The antibodies were expressed after transient transfection in HEK293F cells following the standard protocol as described by the manufacturer (ThermoFisher, Waltham, MA, USA). The antibodies were then purified from cell supernatant using Protein G High Performance Spintrap columns (GE Healthcare, Chicago, IL, USA) as recommended by the manufacturer. The purity and integrity of the proteins were checked by Coomassie stained SDS-PAGE (reducing and non-reducing) and SE-HPLC.

### Binding of monoclonal antibodies to BKPyV VP1 pentamer

The binding of monoclonal antibodies to BKPyV VP1 serotypes Ib1, II, III and IVc2 was analyzed by ELISA. Briefly, Costar Assay Plate 96 well, half-area high binding plates were coated with 30 µl/well 1 µg/ml BKPyV VP1-Ib1, II, III or IVc2 over weekend at 4 °C. Thereafter, plates were blocked for 1 hour using 5% skim milk powder diluted in PBS. After a wash step with PBS 0.05% Tween20, plates were incubated with 30 µl/well of diluted antibodies (in PBS with 0.5% skim milk powder) for 1 hour. Plates were washed with PBS 0.05% Tween20 and then incubated with secondary antibody (HRP-conjugated goat anti-human IgG) diluted 1:10'000 in PBS with 0.5% skim milk powder for 30 min. Plates were washed with PBS 0.05% Tween20. Reaction was developed using 30 µl/well TMB liquid substrate and development stopped after 2.5 min by addition of 15 µl/well 10% $H_2SO_4$. Absorbance was detected at 450 nm. Data were fitted and apparent EC50 ELISA values determined using three-parameter analysis in GraphPad Prism 8.

### Inhibition of BKPyV-VP1 binding to target cells by monoclonal antibodies

Antibodies under evaluation diluted to concentrations of 22.5 nM, 7.5 nM, 2.5 nM, 0.83 nM and 0.28 nM were incubated with VP1 serotype Ib1 pentamer (15.6 nM) or diluted to concentrations of 12 nM, 6 nM, 3 nM, 1.5 nM, 0.75 nM, 0.38 nM and 0.19 nM and incubated with VP1 pentamer serotype II (125.1 nM), serotype III (22.2 nM) or IVc2 (7.8 nM) in binding buffer (PBS + 0.5% FBS) for 30 minutes at ambient temperature. VP1 pentamer concentrations were experimentally determined and are dependent on the affinity of the different serotypes to HEK293TT cells. HEK293TT cells were washed once in binding buffer and resuspended in antibody-VP1-mix or VP1 without antibody at $0.1 \times 10^6$ cells per tested condition. After 1 hour of incubation at 4 °C and two subsequent washing steps, cells were incubated with an AlexaFluor 488 labelled, non-competing anti-VP1 antibody (clone 45F07; Memo Therapeutics AG) at 1:300 dilution in binding buffer for 20 minutes at 4 °C. After two washing steps, cells were analyzed in analytic flow cytometry. Data was analyzed (n = 1, 15000 cells analyzed per concentration) and figure was prepared using FlowJo 10.7.1.

 

## Preparation of BKPyV Pseudoviruses

Pseudovirus were produced following instructions described by Buck et al. [31–37] with slight adaptation. In brief, HEK293TT cells [38] were transfected using a mixture of four plasmids coding for VP1 protein (plasmids pIaw, pII, pIII and pIVc2w encoding VP1 serotype Ia/Ib1, II, III, and IVc2, respectively), VP2 protein (ph2b), VP3 Protein (ph3b) and for Luciferase (secNluc). After overnight incubation, media was exchanged and cells were kept at 37 °C, 5% $CO_2$ for 2 days. Cells were harvested and washed before they were lysed and virus particles were matured by sequentially adding 2 U/ml Neuraminidase, 50 mM Tris pH 8, 0.05% (v/v) Triton X-100 and RNase Cocktail Enzyme Mix (1:1000 dilution, Thermo Fisher Scientific). After a final addition of 25 mM ammonium sulfate pH 9.0 to the cell lysate, final capsid maturation was performed for 23–24 hours at 37 °C. After this overnight incubation, cell lysate was centrifuged and washed and viral particles in the supernatants were kept. After one freeze-thaw cycle of the cell pellet and one additional wash, all supernatants were pooled. Viral particles in the pooled supernatants were purified from free luciferase using gel filtration (2% BCL Agarose Bead Standard (50–150μm), Agarose Beads Technologies). Fractions were tested for presence of infectious viral particle by addition to cells. Only fractions that showed a luciferase signal that could be neutralised by addition of neutralising antibody were pooled.

To determine the titer of the generated virus stocks in form of TCID50/ml, virus stocks were serially diluted and each virus dilution tested in six dependant replicates on a 96-well plate. The virus dilution resulting in 50% of infected wells (i.e., luciferase signal above background, i.e., virus without cells) was determined according to Reed & Muench [39] and TCID50/ml was calculated [40].

## Neutralization of BKPyV pseudovirus

In 96-well cell culture plates HEK293TT were seeded (20 000 cells/well) and left to adhere for 4.75-7.6 h. Serial 12-point antibody dilutions as well as BKPy pseudovirus (final MOI 0.0358) were prepared in cell medium. Antibody and BKPy pseudovirus (serotypes Ia/Ib1, II, III and IVc2) were mixed and incubated for 1 h prior addition to the cells in triplicate (except for 24C03 which was done in duplicate). The mix was incubated for 3 days after which the amount of secreted luciferase in the supernatant was determined by the NanoGlo Luciferase Assay. For this, 40 μl of cell supernatant was transferred to a white, flat bottom 96-well plate and mixed with 40 μl of Nano-Glo Luciferase Assay Buffer and Substrate mix. After incubation at ambient temperature for 2 min, bioluminescence was analysed using the Tecan Spark plate reader. Numerical readout of luminescence counts/s was done with an integration time of 1000 ms and automatic attenuation mode after 3 s linear shaking with 1 mm amplitude (1440 rpm). This allows for a quantitative analysis of infected cells and the determination of the antibody concentration at which all viral particles could be neutralized (no luciferase release). The percentage of infectivity normalized based on the luciferase readout containing BKPy pseudovirus in the presence (100% infection) or absence (0%) of cells. Antibody concentrations were transformed (X = Log(X)) and plotted against the normalized signal and the half maximal inhibitory concentrations (IC50) determined using the "log(Inhibitor) vs. normalized response -- Variable slope" fitting model of GraphPad Prism 10.

## Crystallization of 319C07 Fab fragment with VP1 pentamer

The crystal structure of BKPyV serotype Ib1 VP1 pentamer in complex with 319C07 Fab fragments was determined. Fab fragments were generated digesting 319C07 IgG with papain, using Pierce Fab Preparation Kit (Thermo Fisher, cat no 44985). For complexation, VP1 was mixed with 319C07 Fab fragment in a 1:1.7 ratio. The complex was separated from non-complexed protein by size exclusion chromatography and concentrated to 26 mg/ml. Crystals were grown by hanging drop vapor diffusion method at 20°C, mixing 0.5 μL of the complex with 0.5 μL of reservoir solution containing 21% (w/v) PEG1500, and 0.1 M MIB buffer pH 4.75. The structure has been deposited in the RCSB Protein Data Bank (PDB 9RM2).

Crystallization was performed by Proteros, Munich, Germany as a commercial service.

## Long-term co-culture of replication competent BKPyV on human renal proximal tubular epithelial cells and antibody 319C07

The neutralizing efficacy of antibody mAb 319C07 against BK polyomavirus (BKPyV serotype I strain VR-837) was evaluated using long-term co-cultures with human renal proximal tubular epithelial cells (HRPTEC). HRPTEC were seeded at a density of 3000 cells/well in 96-well plates a day in advance. At assay initiation, medium was refreshed, and antibody dilutions were added at final concentrations ranging from 15 pM to 1.5 nM. BKPyV was introduced at a final dilution of 1:1000. After incubation for one week, viral replication was assessed by qPCR [41] from cell supernatants subjected to viral inactivation by boiling. Remaining cells underwent freeze-thaw cycles to release mature viral particles for subsequent infection cycles. Reinfection and antibody treatment occurred every two weeks for a total of five to six cycles. Viral genome copy numbers were quantified after DNase treatment via qPCR [41] using a standard curve, and potential viral escape mutants were assessed by sequencing the VP1 gene following the final culture passage.

## Supporting information

**S1 Fig. Binding of BKPyV VP1 serotypes II, III and IV to HEK293TT cells in the presence of mAb 319C07.** Attachment of VP1 serotypes II, III and IV pentamer to HEK293TT cells in the absence (dark green) or presence of antibody (mAb319C07, mAb P8D11 or mAb 24C03) at different concentrations (shades of blue) was analyzed. VP1 binding to HEK293TT cells was detected using an AF488 labelled anti-VP1 antibody and compared to cells with detection antibody only (black). Analysis was done by analytic flow cytometry. A shift of the HEK293TT population to higher signals (right-shift) over background (black) indicates binding of VP1 to the cells.
(TIFF)

**S1 Table. Percent binding of BK VP1 serotype II-IV to HEK293TT cells in the presence or absence of antibodies.** Median fluorescent intensity of plots shown in S1 Fig were extracted using FlowJo 10.7.1 and normalized to 100% VP1 binding (VP1 II, III or IV binding to HEK293TT cells in the absence of antibodies) and 0% VP1 binding (HEK293TT cells with detection antibody only). For mAb 319C07 and mAb P8D11, a concentration-dependent decrease in VP1 binding to the cells was observed. mAb P8D11 shows a weaker inhibition of VP1 binding to cells than mAb 319C07 for all VP1 serotypes tested. mAb 24C03 does not inhibit VP1 binding to HEK293TT cells. n = 1, 15000 cells analyzed per concentration.
(TIF)

**S2 Fig. Long term culture of replication competent BKPyV STIV in the presence of antibody mAb 319C07 and subsequent analysis of putative VP1 sequence alterations.** A) Neutralization capacity of mAb 319C07 and reference antibody mAb P8D11 plus negative control mAb 24C03 on HRPTEC infected with BKPyV serotype IV was quantified. With antibodies mAb 319C07 and reference antibody mAb P8D11 assessment was performed in duplicate at all concentrations, for mAb 24C03, the highest concentration was used. Plotted are mean and ranges of n = 2. B) Sequence analysis performed with the 4th infection passage showed no occurrence of mutated virus.
(TIF)

**S1 Supplementary Method. Long-term co-culture of replication competent BKPyV STIV on human renal proximal tubular epithelial cells and antibody 319C07** .
(DOCX)

**S1 Data. Raw data for Fig 2.**
(XLSX)

**S2 Data. Raw data for Fig 3 And Table 2. Compressed Folder contains plate layout, fcs files, FlowJo evaluation and extracted median fluorescence used to prepare Table 2.**
(ZIP)

**S3 Data. Raw data for Fig 4.**
(XLSX)

**S4 Data. Raw data for Table 3. Summarizes IC50 values of individual neutralization experiments.**
(XLSX)

**S5 Data. Raw data for Fig 5.**
(XLSX)

**S6 Data. Raw data for Fig 7.**
(XLSX)

**S7 Data. Raw data for S1 Fig and S1 Table.**
(XLSX)

**S8 Data. Raw data for S2 Fig.**
(XLSX)

## Acknowledgments

The authors want to thank the blood donors who donated blood from which the B cell repertoire libraries were made, as well as Dr. Chris Buck for providing BKPyV pseudovirus plasmids and HEK293TT cells. We also want to thank Dr. Alba Lepore for expert help with the analysis of the structural data. Flow cytometry was performed at the flow cytometry facility of the University of Zurich.

## Author contributions

**Conceptualization:** Marcel Weber, Simone Schmitt, Thomas Schachtner, Serena Delbue, Armin Mäder, Christoph Esslinger, Matthias Hillenbrand.

**Data curation:** Matthias Hillenbrand.

**Formal analysis:** Marcel Weber, Barbara Eicher, Jemima Seidenberg, Justina Rutkauskaite, Benedikt Stöckli, Catherine Townsend, Matthias Hillenbrand.

**Funding acquisition:** Christoph Esslinger.

**Investigation:** Marcel Weber, Simone Schmitt, Barbara Eicher, Jemima Seidenberg.

**Methodology:** Marcel Weber, Simone Schmitt, Barbara Eicher, Jemima Seidenberg, Justina Rutkauskaite, Benedikt Stöckli, Catherine Townsend, Uyen Huynh-Do, Serena Delbue, Matthias Hillenbrand.

**Project administration:** Marcel Weber, Simone Schmitt, Christoph Esslinger, Matthias Hillenbrand.

**Supervision:** Simone Schmitt, Christoph Esslinger, Matthias Hillenbrand.

**Visualization:** Barbara Eicher, Jemima Seidenberg.

**Writing – original draft:** Simone Schmitt, Christoph Esslinger, Matthias Hillenbrand.

**Writing – review & editing:** Simone Schmitt, Barbara Eicher, Christoph Esslinger, Matthias Hillenbrand.

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
