## [Decision Letter · Decision Letter 0]

PPATHOGENS-D-25-00807

A Highly Potent Human Antibody Neutralizing All Serotypes of BK Polyomavirus

PLOS Pathogens

Dear Dr. Hillenbrand,

Thank you for submitting your manuscript to PLOS Pathogens. After careful consideration, we feel that it has merit but does not fully meet PLOS Pathogens's publication criteria as it currently stands. Therefore, we invite you to submit a revised version of the manuscript that addresses the points raised during the review process.

Please submit your revised manuscript within 30 days Jul 15 2025 11:59PM. If you will need more time than this to complete your revisions, please reply to this message or contact the journal office at plospathogens@plos.org. Please include the following items when submitting your revised manuscript:

We look forward to receiving your revised manuscript.

Kind regards,

Walter J. Atwood

Academic Editor

PLOS Pathogens

Alison McBride

Section Editor

PLOS Pathogens

Sumita Bhaduri-McIntosh

Editor-in-Chief

PLOS Pathogens

orcid.org/0000-0003-2946-9497

Michael Malim

Editor-in-Chief

PLOS Pathogens

orcid.org/0000-0002-7699-2064

**Journal Requirements:**

At this stage, the following Authors/Authors require contributions: Christoph Esslinger, and Simone Schmitt. Please ensure that the full contributions of each author are acknowledged in the "Add/Edit/Remove Authors" section of our submission form.

https://journals.plos.org/plospathogens/s/submission-guidelines#loc-parts-of-a-submission

3) We noticed that you used the phrase 'data not shown' in the manuscript. We do not allow these references, as the PLOS data access policy requires that all data be either published with the manuscript or made available in a publicly accessible database. Please amend the supplementary material to include the referenced data or remove the references.

4) We do not publish any copyright or trademark symbols that usually accompany proprietary names, eg ©,  ®, or TM  (e.g. next to drug or reagent names). Therefore please remove all instances of trademark/copyright symbols throughout the text, including:

- ® on pages: 3, 4, 13, 15, and 16.

5) Please upload all main figures as separate Figure files in .tif or .eps format. For more information about how to convert and format your figure files please see our guidelines: 

6) We notice that your supplementary Figures, and Tables are included in the manuscript file. Please remove them and upload them with the file type 'Supporting Information'. Please ensure that each Supporting Information file has a legend listed in the manuscript after the references list.

7) We are unable to open the following Supporting Information file: OneDrive_2_14.04.2025.zip. Please kindly revise as necessary and re-upload.

8) We note that your Data Availability Statement is currently as follows: "we are consenting to make all data available as stated in the PLOS Policy". Please confirm at this time whether or not your submission contains all raw data required to replicate the results of your study. Authors must share the “minimal data set” for their submission. PLOS defines the minimal data set to consist of the data required to replicate all study findings reported in the article, as well as related metadata and methods (https://journals.plos.org/plosone/s/data-availability#loc-minimal-data-set-definition).

- The points extracted from images for analysis..

9) Please amend your detailed Financial Disclosure statement. This is published with the article. It must therefore be completed in full sentences and contain the exact wording you wish to be published. Please ensure that the funders and grant numbers match between the Financial Disclosure field and the Funding Information tab in your submission form. Note that the funders must be provided in the same order in both places as well.

**Reviewers' Comments:**

Reviewer's Responses to Questions

**Part I - Summary**

Reviewer #1: This study presents the discovery of several monoclonal antibodies that can neutralize the BK polyomavirus. In particular, one of these antibodies (named 319C07) is shown to have especially potent activity against all four BKPyV-

Serotypes. This antibody was isolated from a cognate memory B cell repertoire

library obtained from peripheral blood of a kidney transplant recipient.

The authors show that mAb 319C07 has somewhat higher affinity for the BKPyV genotypes I-IV compared with a previously identified antibody P8D11. However, the increased potency ranges between 2-fold and 3-fold higher affinity for genotypes I, II and IV, with no difference in affinity in genotype III. The new antibody is no doubt highly potent, but in my opinion the manuscript’s writing style (discovery of an “ultra-potent” antibody, etc.) raises expectations that are not quite matched by the data shown in Figure 2 and Table 1. Likewise, the inhibition of binding of BKPyV VP1 to HEK283TT cells (Figure 3) shows a small improvement compared to P8D11.

The new antibody is however much more effective in neutralizing all four genotypes of BKPyV compared to previously identified antibodies. Therefore, there are good reasons to expect that the antibody can be helpful for developing new strategies for the treatment of BKPyV infections in renal transplant recipients.

The manuscript includes a structural characterization of the complex of the Fab region of antibody 319C07 and the VP1 pentamer of BKPyV (genotype I). The crystal structure analysis was done commercially, and there is a report on the structure determination included in the supporting information. The structure has high resolution, and good statistics. The structure shows that five Fabs bind to the top of the VP1 pentamer, and the authors state that the antibody binds to the highly conserved amino acid stretch N273–T277, which is important for sialic acid binding. I found this section of the manuscript somewhat lacking in detail – it would be helpful I think to include a detailed figure showing the detailed contacts for one Fab, with specific residues. What contacts are formed? Do the interactions preclude binding of (i.e. clash with) the receptor? What is the buried surface area in the complex?

Maybe one could show an overlay of the structures bound to the receptor and to the Fab? I think such a more detailed figure of the interactions would be very helpful.

Furthermore, it would also be helpful to highlight residues that differ among the BK genotypes and their role(s) in contact formation, in order to illustrate how the antibody is able to recognize all 4 genotypes. I noted that the antibody 319C07 does not neutralize genotype III nearly as efficiently as the other three genotypes (Table III). Can this be explained by differences in the sequence (specific residues that are different) in the Fab-binding region?

Reviewer #2: In this study, Weber and colleagues characterize a highly potent BKPyV-neutralizing human mAb.

This is excellent work with high clinical relevance for an important disease state. All my suggestions are minor – primarily aimed at improving clarity.

**Part II – Major Issues: Key Experiments Required for Acceptance**

Reviewer #1: I think the manuscript would benefit from the inclusion of affinity data (e.g. surface plasmon resonance), although I don’t see this as a requirement for acceptance.

Reviewer #2: I do not find any major issues and I do not think new experiments are required

**Part III – Minor Issues: Editorial and Data Presentation Modifications**

Reviewer #1: Minor comments:

• Figure 6 needs labels, maybe also highlight one VP1 monomer to show how the Fabs bridge two VP1 monomers.

• The authors write in the discussion that “the joining site between VP1 monomers is highly conserved, and mutations in this region to avoid binding of mAb 319C07 would likely disrupt capsid formation, significantly compromising viral fitness”. I am not sure I follow the reasoning here. I would think that the Fab does not directly bind to residues that are involved in VP1-VP1 interactions within the pentamer, but rather to surface-exposed residues in this area, correct? Thus, mutations of such surface-exposed residues would likely be tolerated by the capsid.

Reviewer #2: •Intro sentence 2: it would be more informative (and less redundant) to say “forms a non-enveloped capsid.”

•Intro sentence 3: BKPyV-IV doesn’t require gangliosides for entry

https://pubmed.ncbi.nlm.nih.gov/23843634/

Instead of “engaging gangliosides in the cell membrane” it would be more accurate to say “engaging sialylated glycans.”

•The last sentence of the first Intro paragraph is missing a reference. The sentence could simply be deleted. If it’s kept, it would also be important to note that bladder epithelial cells are a typical productive target tissue for BKPyV.

•Bottom of page 2: cite NCT04294472

•Top of page 3: consider removing the comma in the phrase “recipients, who” - the comma confusingly seems to imply that all kidney transplant recipients demonstrate fast clearance.

•Methods: I’m not finding details on how the VP1 pentamers produced? What was the fluorochrome? Which exact strain was used? It’s puzzling that the manuscript alternates between describing it as Ia and Ib1 because the Ia/Ib1 distinction is based on sequences outside the VP1 ORF (i.e., Ia and Ib1 genotypes are indistinguishable at the level of VP1 protein sequence). Is it possible it was actually a Ib2 strain? This is important because Ib2 has been proposed as a disproportionate culprit behind PVAN in some world regions.

•bottom of page 14: typo – “clones cloned”

•Bottom pf page 17: which BKPyV strain was used for the HRPTEC culture experiment?

•Bottom of page 4: “10,000 IU/mL” could leave readers with the false impression that Infectious Units were measured. It appears that something more like “copies/ml” is being measured. Also note: “viremia” may not be correct – the Hirsch lab recently presented a compelling case for the idea that the measurement would more accurately be described as “DNAemia” https://pubmed.ncbi.nlm.nih.gov/31759262/

•Page 7 (cell attachment results): The statement “These results suggest that the neutralizing effect of both antibodies is likely mediated via blockade of viral attachment to cells” is an over-extrapolation of the data. The results convincingly show that 319C07 prevents the pentamer from engaging sialylated glycans on the cell surface. Although this effect would indeed be neutralizing in many contexts, it’s important to note that the assembled virion also has binding pockets that can engage glycosaminoglycans. https://pubmed.ncbi.nlm.nih.gov/29091757/

The GAG-binding pockets are absent from free pentamers. It thus remains conceivable that a full-size virion could bind the cell surface via engagement of cell-surface GAGs even while 319C07 is blocking the engagement of sialylated glycans. The text should be re-written to focus on the results indicating specific interference with the sialylated glycan-binding step in the infectious entry process, as opposed to all possible forms of cell attachment. There's a similar problem at the top of page 10 – BKPyV has two distinct glycan receptors/co-receptors. It’s important to be specific about the fact that the mAb is blocking the sialylated glycan binding pocket.

•Page 11 (long term propagation experiment): since BKPyV is a DNA virus replicated with high fidelity by host cell DNA polymerases, it generally accumulates mutations at a very slow rate. The lack of observed mutations is thus consistent with general theory. On the other hand, it appears that the viral genome can be acutely targeted by host cell APOBEC3B: https://pubmed.ncbi.nlm.nih.gov/29746834/

HRPTEC cells may not express much APOBEC3B in culture. It would be useful for the Discussion section to mention the idea of testing 319C07 against stereotypical APOBEC3B-signature mutations that are typically found in PyVAN cases. It could be framed as a limitation of the current study that might be addressed in future work.

PLOS authors have the option to publish the peer review history of their article (what does this mean? ). If published, this will include your full peer review and any attached files.

**Do you want your identity to be public for this peer review?** For information about this choice, including consent withdrawal, please see our Privacy Policy .

Reviewer #1: No

Reviewer #2: **Yes: ** Chris Buck

**Figure resubmission:**
---

## [Editor Report · Decision Letter 1]

Dear Dr. Hillenbrand,

We are pleased to inform you that your manuscript 'A Highly Potent Human Antibody Neutralizing All Serotypes of BK Polyomavirus' has been provisionally accepted for publication in PLOS Pathogens.

Best regards,

Walter J. Atwood

Academic Editor

PLOS Pathogens

Alison McBride

Section Editor

PLOS Pathogens

Sumita Bhaduri-McIntosh

Editor-in-Chief

PLOS Pathogens

orcid.org/0000-0003-2946-9497

Michael Malim

Editor-in-Chief

PLOS Pathogens

orcid.org/0000-0002-7699-2064
---

## [Editor Report · Acceptance letter]

Dear Mr Hillenbrand,

We are delighted to inform you that your manuscript, " A Highly Potent Human Antibody Neutralizing All Serotypes of BK Polyomavirus ," has been formally accepted for publication in PLOS Pathogens.

Best regards,

Sumita Bhaduri-McIntosh

Editor-in-Chief

PLOS Pathogens

orcid.org/0000-0003-2946-9497

Michael Malim

Editor-in-Chief

PLOS Pathogens

orcid.org/0000-0002-7699-2064